# Changes within the central stalk of *E. coli* $F_1F_o$ ATP synthase observed after addition of ATP

Meghna Sobti[1,2], Yi C. Zeng [1,2], James L. Walshe[1], Simon H. J. Brown[3], Robert Ishmukhametov[4] & Alastair G. Stewart [1,2✉]

$F_1F_o$ ATP synthase functions as a biological generator and makes a major contribution to cellular energy production. Proton flow generates rotation in the $F_o$ motor that is transferred to the $F_1$ motor to catalyze ATP production, with flexible $F_1/F_o$ coupling required for efficient catalysis. $F_1F_o$ ATP synthase can also operate in reverse, hydrolyzing ATP and pumping protons, and in bacteria this function can be regulated by an inhibitory $\varepsilon$ subunit. Here we present cryo-EM data showing *E. coli* $F_1F_o$ ATP synthase in different rotational and inhibited sub-states, observed following incubation with 10 mM MgATP. Our structures demonstrate how structural transitions within the inhibitory $\varepsilon$ subunit induce torsional movement in the central stalk, thereby enabling its rotation within the $F_o$ motor. This highlights the importance of the central rotor for flexible coupling of the $F_1$ and $F_o$ motors and provides further insight into the regulatory mechanism mediated by subunit $\varepsilon$.

[1] Molecular, Structural and Computational Biology Division, The Victor Chang Cardiac Research Institute, Darlinghurst, NSW, Australia. [2] School of Clinical Medicine, Faculty of Medicine and Health, UNSW Sydney, Sydney, NSW, Australia. [3] Molecular Horizons, University of Wollongong, and Illawarra Health and Medical Research Institute, Wollongong, NSW, Australia. [4] Department of Physics, Clarendon Laboratory, University of Oxford, Oxford, UK. ✉email: a.stewart@victorchang.edu.au

A key component in the generation of cellular energy is the $F_1F_o$ ATP synthase, a biological rotary motor that converts proton motive force (pmf) to adenosine triphosphate (ATP) in both oxidative phosphorylation and photophosphorylation[1–4]. The enzyme is based on two rotary motors, termed $F_1$ and $F_o$, that are coupled together by two stalks: a central rotor stalk and a peripheral stator stalk (Fig. 1a). The central stalk is comprised of subunits ε and γ and rotates during the enzyme's catalytic cycle. The peripheral stalk is a homodimer comprised of b subunits, and connects the non-rotating components together. During ATP synthesis, the membrane-bound $F_o$ motor converts the pmf into rotation of the $F_o$ rotor ring, this rotation is transferred to the $F_1$ motor via the central rotor which in turn induces conformational changes in the catalytic subunits. These conformational changes alter the chemical environment in the catalytic sites within $F_1$ so that ATP is synthesized from ADP and inorganic phosphate ($P_i$)[5,6]. The $F_1$ motor has pseudo three-fold symmetry[5,7] with six dwell positions[8], but the symmetry of the $F_o$ varies across species with the *Escherichia coli* enzyme having tenfold rotational symmetry (10 c subunits in the rotor ring[9]). Because of the symmetry mismatch between the $F_1$ and $F_o$ motors, it has been hypothesized that they need to be flexibly coupled for efficient function[10], with the peripheral and/or central stalks flexing to alleviate the symmetry mismatch between them[11,12].

$F_1F_o$ ATP synthase can operate in reverse, hydrolyzing ATP and pumping protons. However, cells have evolved inhibitory mechanisms to avoid wasteful hydrolysis of ATP that could occur under certain physiological conditions. Bacterial ATP synthases appear to utilize a range of different mechanisms for inhibition, with nucleotides, ions and conformational changes making contributions[13–15]. *E. coli* and other related bacteria exploit an internal subunit, termed ε, that can sense the environment and inhibit the enzyme under certain conditions[13]. In *E. coli*, subunit ε can be divided into N-terminal (εNTD) and C-terminal (εCTD) domains. The εNTD is a β-sandwich with two five-stranded sheets whereas the εCTD is formed from two short helices, residues 87–102 and 110–136 (referred to as εCTH1 and εCTH2 respectively), connected by a linker (Fig. 1b). It has been hypothesized that the εCTD is able to inhibit the $F_1$ motor by extending up and blocking rotation under certain conditions[16,17]. Multiple structural studies examining *E. coli* $F_1$ or $F_1F_o$ ATP synthase either in the absence of nucleotide[18] or in the presence of AMPPNP[16] or MgADP[19], have shown the εCTD oriented in an extended up position (Fig. 1b), whereas the isolated ε subunit has also been crystallized in a condensed down position[20] (Fig. 1c). Previously, we reported ~5 Å resolution structure of *E. coli* $F_1F_o$ ATP synthase following incubation with 10 mM MgATP[21], which confirmed that the εCTD transitions to a condensed down conformation via a half-up intermediate.

To define the transitions of the εCTD in greater detail and understand how it regulates function, we have used cryo-EM to examine detergent-solubilized *E. coli* $F_1F_o$ ATP synthase[22] following a 45 s incubation with 10 mM MgATP. These conditions allow the enzyme to be observed operating in the hydrolysis direction, and are similar to the concentrations found in *E. coli* undergoing aerobic respiration[23] (though with much lower Pi concentration[24]). Strikingly, the improved detail in this study shows that the transition of the εCTD to a down sub-state is associated with a substantial torsional flexing of the central stalk, as evident by rotation of the εNTD about the γ subunit. This flexing, in combination with bending in the peripheral stalk, mediates rotation in the $F_o$ motor. Our work provides a structural framework for role of the central stalk in flexible coupling within $F_1F_o$ ATP synthases and suggests a hypothesis of how pmf could modulate the enzyme's structure and function.

## Results

**Cryo-EM analysis of $F_1F_o$ ATP synthase following incubation with MgATP.** Cryo-EM maps of *E. coli* $F_1F_o$ ATP synthase in the presence of 10 mM MgATP were obtained using methods similar to those in previous studies[18,19,21,25] (Supplementary Figs. 1 and 2) and provided superior structural information than was observed previously[21]. Previous work[18] had identified three major conformational states of the enzyme in which the central stalk is rotated by ~120° relative to peripheral stalk. These states are termed State 1, State 2 and State 3, with the order referring to the enzyme operating in ATP hydrolysis direction (Supplementary Fig. 2). An overall resolution of 2.7–3.0 Å was achieved for the three rotational states, which enabled bound nucleotides to be identified and modeled (Fig. 2a). The three catalytic β subunits in the present study were identified as containing β1 ($β_{DP}$); MgADP, β2 ($β_E$); ADP and β3 ($β_{TP}$); MgATP (Fig. 2a). Compared to the same enzyme imaged in the presence of 10 mM MgADP[19], the central stalk had rotated as a rigid body by ~10° counterclockwise, when viewed from the membrane (Fig. 2b and Supplementary Fig. 3), the εCTH2 has dissociated from the central stalk, and the β1 ($β_{DP}$) subunit has closed to contact the γ subunit (Fig. 2c) and bind MgADP. Comparison of the relative position of the rotor axel between known structures[8,16,19,26–28], highlighted that the $F_1$-ATPase was in a similar rotary position to that of observed of *Geobacillus stearothermophilus* (also termed *Bacillus* PS3) in the catalytic dwell[8] (Supplementary Fig. 3), indicating that this enzyme was in a similar state. In the maps of these three states, the details of the $F_o$ region remained ambiguous and hence further data processing was performed to verify the location of the εCTD and c-ring, as described in the following section.

Masked 3D classification focusing on the central rotor highlighted sub-states in which the εCTD adopted either a

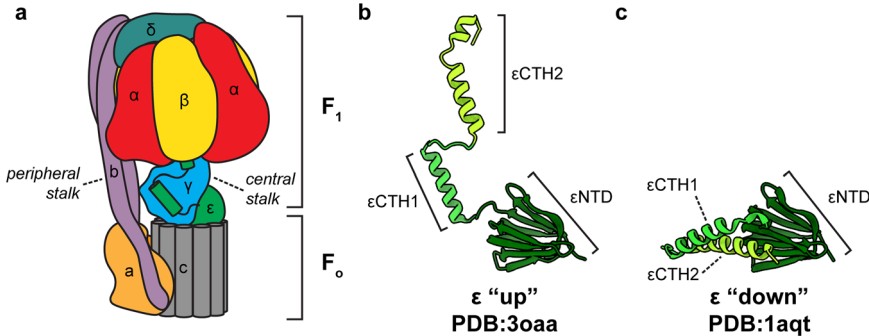

**Fig. 1 The ε subunit of *E. coli* $F_1F_o$ ATP synthase. a** Schematic of *E. coli* $F_1F_o$ ATP synthase with subunits colored and labeled. The ε subunit is shown in green in the inhibited up conformation. **b** Crystal structure of the ε subunit in the up conformation (PDB:3oaa[16]), with α, β, and γ subunits removed for clarity. **c** Crystal structure of the isolated ε subunit in the down conformation (PDB:1aqt[20]).

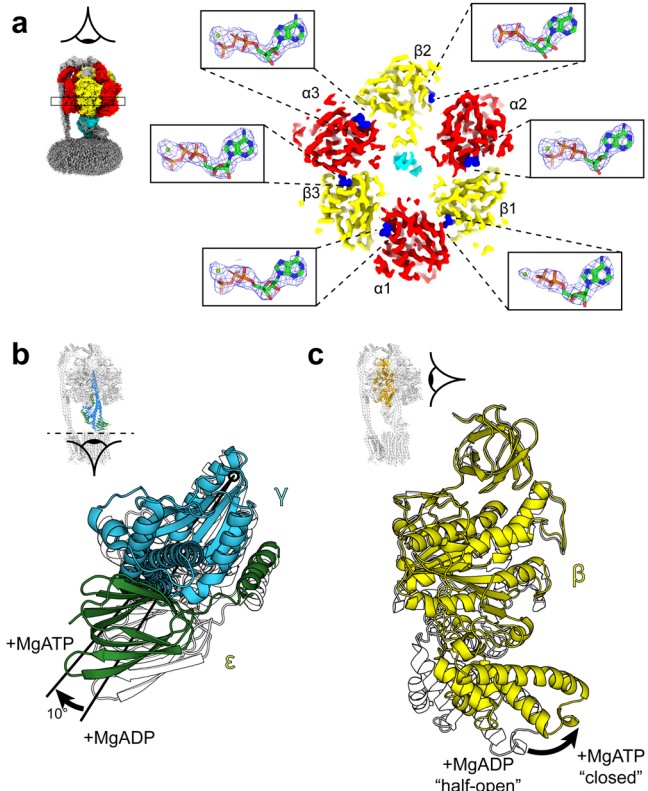

**Fig. 2 Nucleotide occupancy and conformational changes in the F₁ motor following incubation with MgATP. a** Horizontal section of the State 2 *E. coli* F₁Fₒ ATP synthase cryo-EM map and details of the β subunit catalytic site occupancies (with equivalent mitochondrial F₁ nomenclature:[5] β1 = β_DP, β2 = β_E, β3 = β_TP, as named for the *E. coli* enzyme[16]). β1 contains MgADP, β2 contains ADP, and β3 contains MgATP. Section of map contoured to 0.028 in ChimeraX[62] and mesh for nucleotides contoured to isolevel 10 in PyMol (Schrödinger). **b, c** Comparison of the F₁ motor after incubation with MgATP (this study; γ in blue, ε in green and β in yellow) or MgADP (PDB:6oqv[19]; subunits shown as outline). **b** The central stalk (subunits γ and ε) is rotated ~10° clockwise when viewed from the membrane (structures are aligned to the F₁ β barrel crown). **c** β1 closes inwards from a half-open to closed conformation.

condensed down conformation or an extended half-up conformation (Fig. 3 and Supplementary Fig. 2). The maps generated for each of these sub-states showed the position of the εCTD, with local resolution estimates of 4–5 Å (Supplementary Fig. 4). However, due to the likely high flexibility of this sample, it was still difficult to unequivocally assign the position of the membrane domain subunits (a, b, and c subunits) in the maps. To obtain clearer information in this region, a refinement that focused on the Fₒ motor was performed (Supplementary Fig. 2) and produced maps of sufficient detail to enable fitting of the membrane region. These maps were combined (using Phenix[29]; maps and models provided as Supplementary Data 1–4) with the maps obtained without focused refinement to visualize overall changes between sub-states.

**Subunit ε stabilizes the conformation of the central stalk.** In this study we observed the εCTD in two conformations, a half-up and a down sub-state. Although the resolution of the maps was not atomic (Fig. 3 and Supplementary Fig. 4) they contained clear detail (4-5 Å resolution) that showed the position of the α-helices and enabled the docking and fitting of known crystal[20] and cryo-EM[19] models (Supplementary Fig. 5), providing a clear picture of

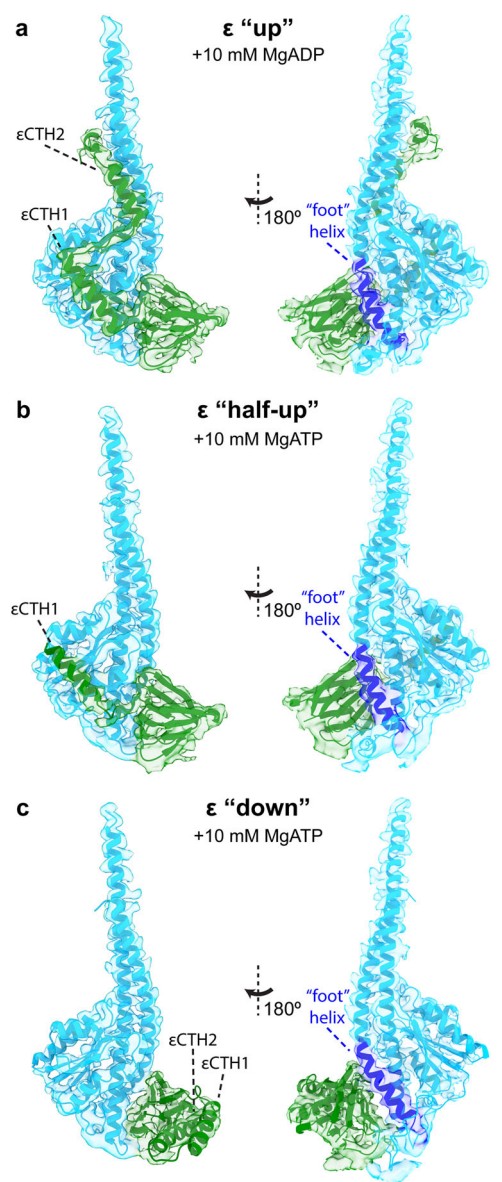

**Fig. 3 *E. coli* F₁Fₒ ATP synthase ε subunit in three conformational sub-states.** Cryo-EM maps (transparent surface) and molecular models (cartoon representation) of the *E. coli* F₁Fₒ ATP synthase rotor in three conformation sub-states. Subunit γ in light blue and ε in green, with the foot helix of subunit γ labeled in dark blue. **a** The εCTD up sub-state observed after addition of 10 mM MgADP (PDB:6oqv; EMDB:20171[19]). **b** The εCTD half-up sub-state observed after addition of 10 mM MgATP (State 2 half-up in this study). **c** The εCTD down sub-state observed after addition of 10 mM MgATP (State 2 down in this study). See Supplementary Fig. 5 for close up views of the cryo-EM maps for the εCTD and γ foot.

the molecular arrangement in this system. In the half up sub-states, no density for εCTH2 was observed, however εCTH1 remained attached to subunit γ (Fig. 3b). In the down sub-state, both the εCTH1 and εCTH2 were folded on to each other as seen in the isolated *E. coli* ε subunit crystal structure[20] (Fig. 3c). The sub-states of State 2 showed the best detail for the two εCTD conformations (Fig. 3 and Supplementary Fig. 4) and allowed the c subunits to be assigned to Fₒ rotational sub-states based on their interaction with the εNTD (Supplementary Fig. 6), assuming that the c-ring remains bound during rotation, which is likely because the interface between the ring and central rotor has a buried surface of ~1200 Å²[30].

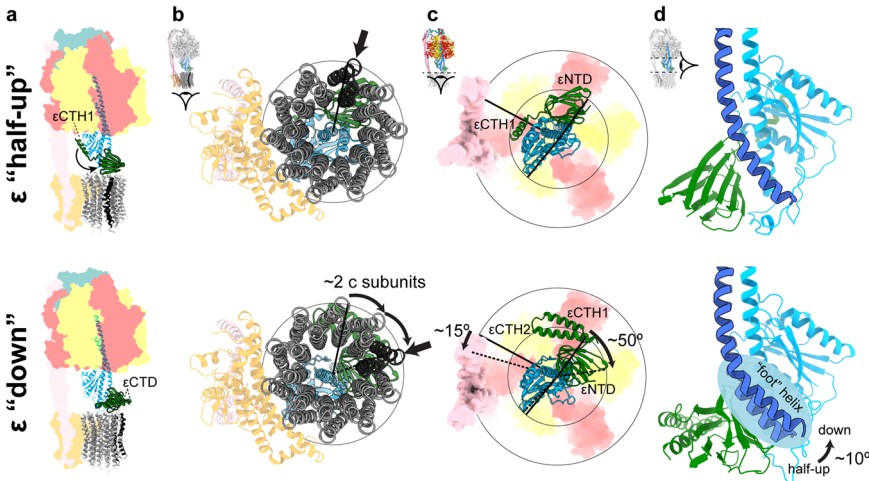

**Fig. 4 Structural rearrangements of E. coli F₁Fₒ ATP synthase between the half-up and down conformations.** Comparison of State 2 εCTD half-up conformation (top panels) with State 2 εCTD down conformation (bottom panels). **a** Overall view describing the εCTD (green) transition and rotation of the c-ring (gray with one c subunit colored black). **b** When superposed on subunit a (orange), the c-ring rotates two c subunits in the Fₒ motor (one c subunit colored black with black arrow—relative rotation of c subunits identified using the interaction between εNTD and c subunit; Supplementary Fig. 6). **c** When superposed on the β barrel crown of the α and β subunits, the εNTD rotates ~50° about the γ subunit and the peripheral stalk flexes ~15°. **d** When superposed on N-terminus of subunit γ, the foot helix (residues γ39-57) bends and twists to accommodate the movement of the εNTD.

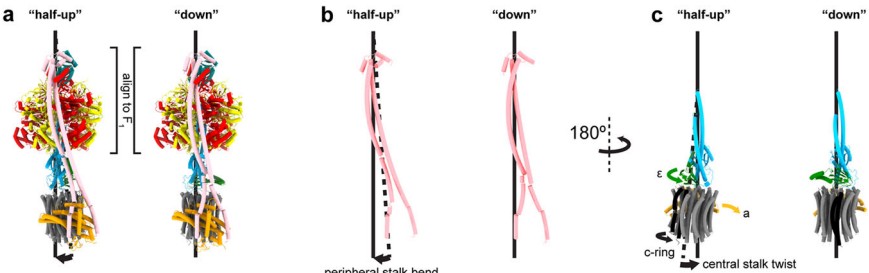

**Fig. 5 Movements of the central and peripheral stalks.** Side views showing the comparison of State 2 half-up and down sub-states. **a** Intact F₁Fₒ complexes shown as tube cartoon. **b** The peripheral stalk (dimer of b subunits) bends counterclockwise and **c** the central stalk twists clockwise, facilitating a rotation of two c subunits in the Fₒ motor.

When the State 2 εCTD half-up and down structures were compared (Fig. 4) additional clear differences beyond the εCTD up and down conformation were observed. When aligned on the stator a subunit (Fig. 4b), the Fₒ ring rotates the equivalent of two c subunits in a clockwise direction when viewed from the membrane (akin to the synthesis direction). This rotation of the Fₒ motor was facilitated mainly by a twisting of the central stalk (~50°), but also by flexing of the peripheral stalk (~15°) (Fig. 5), with the remainder facilitated by small rearrangements within the complex. The twisting of the central stalk involves the εNTD rotating about the γ subunit (Fig. 4c) and this is mediated by a ~10° bending at the foot of the N-terminal γ subunit helix (defined by residues γ39-57) (Figs. 3c and 4d). When the εCTD is half-up, εCTH1 provides an additional link between the εNTD and subunit γ, stabilizing the conformation of the central stalk together (Figs. 3b and 6). When the εCTD is in the down conformation, the central stalk is no longer stabilized by the εCTH1:γ interaction, likely increasing its torsional freedom, and the εNTD rotates relative to the γ subunit. The rotational movement of the εNTD observed between the half-up and down sub-states increases the distance between the εNTD and εCTH1 binding site on subunit γ (Fig. 6), reducing the likelihood of the εCTH1:γ interaction.

**εCTD helix mutants alter enzyme function.** To further investigate the role of each of the εCTD helices on enzyme function and the impact the half-up sub-state has on ATP hydrolysis, two ε

subunit truncation mutants were generated. An εΔCTH2 mutant (lacking residues ε105–124; Fig. 7a) would lack the ability to form the classical autoinhibited up sub-state[16] or the uninhibited down sub-state, but would retain the ability to form the half-up sub-state and stabilize the central stalk conformation. An εΔCTH1 + 2 mutant (lacking residues ε82–124; Fig. 7b) lacking both εCTD helices would mimic the uninhibited down sub-state and prevent the up or half-up conformations, freeing the central stalk from the bridge made by the εCTH1.

First, to confirm that the εΔCTH2 truncation was still able to form the half-up sub-state, we obtained cryo-EM maps of the εΔCTH2 truncation in the presence of 10 mM MgATP (Fig. 7c and Supplementary Fig. 7). Although processing and masking methods similar to those used on the wild-type enzyme were performed on these data, only a single conformation of the εΔCTH2 subunit was observed. The maps of this sample showed the εCTH1 was bound to the γ subunit in a manner analogous to that observed in the wild-type half-up sub-state. ATP regeneration assays performed on wild-type and truncation mutants showed different turnover rates (Fig. 7d, e). As expected, εΔCTH1 + 2 showed the highest turnover of ATP because it is unable to form either the up or half-up inhibited sub-states. However, εΔCTH2 showed a higher turnover than wild-type enzyme but lower turnover than εΔCTH1 + 2, indicating that the half-up position of εCTH1and its interaction with γ subunit somehow modulates the enzyme turnover.

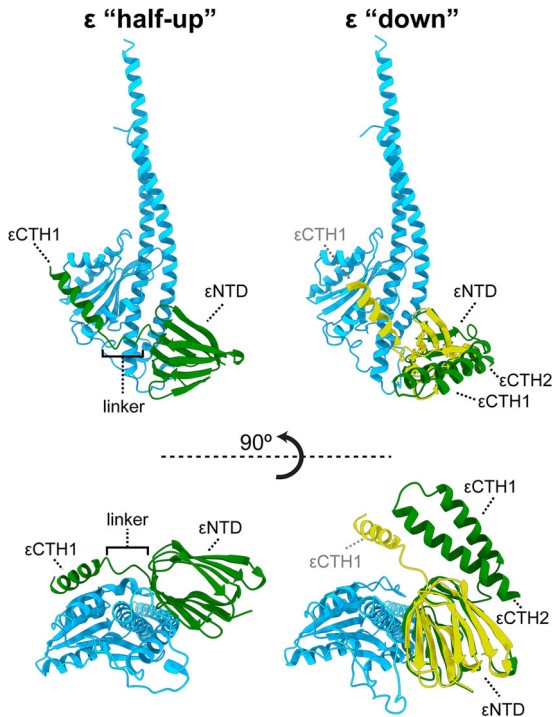

ε "half-up"　　　　　ε "down"

εCTH1　　　　　　　　εCTH1

εNTD　　　　　　　　εNTD

εCTH2
εCTH1

linker

90°

linker　　　εNTD

εCTH1

εCTH1

εCTH1

εCTH2

εNTD

**Fig. 6 Rotation of the εNTD increases the distance to the εCTH1 binding site.** To highlight the relative rotation of the ε subunit between the half-up and down sub-states, the State 2 half-up ε subunit was superposed onto the State 2 down εNTD (shown in yellow). After the εNTD rotates about the γ subunit, the distance between εCTH1 and its binding site on γ is increased in the down sub-state and is less likely to attach to the central stalk.

## Discussion

The cryo-EM and functional studies presented here provide information on how $F_1F_o$ ATP synthase changes conformation after addition of 10 mM MgATP. The central stalk not only transitions to a down state, but also twists, allowing rotation in the $F_o$ motor. This twisting, in combination with flexing in the peripheral stalk, may facilitate flexible coupling between the $F_1$ and $F_o$ motors and suggests a hypothesis of how subunit ε can modulate *E. coli* $F_1F_o$ ATP synthase function.

Although the enzyme is likely rotating and hydrolyzing ATP during the freezing process[21], we did not observe sub steps (e.g., the binding dwell) in the enzyme beyond the three catalytic dwells and ε/c-ring sub rotation. This is likely due to the limited time the enzyme would spend outside the catalytic dwell under these imagining conditions, with single molecule studies needing external load, in the form of increased medium, to observe sub-states[9]. Assuming that the $F_1F_o$ enzyme has similar turnover to the $F_1$-ATPase, single molecule studies suggest that the enzyme would be in the catalytic dwell ~97% of the time[31], and hence too small number of particles for efficient sorting. Further to this, the relative number of particles between each state differed substantially and the proportional differences were different than that observed for the same enzyme incubated with MgADP[19]. Other work using single molecule methods has suggested that the 3:10 symmetry mismatch between the $F_1$ and $F_o$ motors would cause asymmetry in the c-ring rotation[32–34], and the structural data of *E. coli* $F_1F_o$ incubated with either MgADP[19] or MgATP certainly corroborates this.

Flexible coupling between the $F_1$ and $F_o$ motors is necessary to facilitate efficient enzyme function, however whether this flexibility originates from the peripheral or central stalk has been

controversial[35–40]. To date, structural studies have only shown flexibility within the peripheral stalk[12], with the central stalk remaining rigid in all rotational sub-steps observed[19,41–46]. Previously we have shown that, in *E. coli*, the peripheral stalk can flexibly couple the $F_1$ and $F_o$ motors, facilitating a single c subunit step in the $F_o$ motor without flexing of the central stalk[19]. In the present study we observe that the $F_o$ motor rotates two c subunit steps in the clockwise direction (when viewed from the membrane, i.e. the synthesis direction) when the εCTD transitions from the half-up to the down conformation (Figs. 4b and 8a). Torsional flexing of the central stalk facilitates the majority of this $F_o$ rotation, while the peripheral stalk flexes to accommodate the remainder (Fig. 5). In this experiment MgATP was added prior to grid freezing (similar but not identical conditions to that seen in *E. coli* undergoing aerobic respiration) and therefore the enzyme would be rotating in the counterclockwise direction (when viewed from the membrane, i.e. the hydrolysis direction). Hence, the rotation we observe in $F_o$ when the εCTD transitions from the half-up to down conformation is in the opposite direction to that in which $F_o$ is being driven during ATP hydrolysis. The counter rotation we observe here could be due to several factors. Single molecule studies[33,47] have identified rotation in $F_o$ motor of a similar magnitude and direction, suggesting this could be indicative of sub stepping in the synthesis direction. Another possibility could be that that the drag, that would be experienced at the stator/lipid/detergent interfaces, results in the delay of the $F_o$ ring during rotation that we observe here (Fig. 8a).

Although single molecule[36,47,48] and molecular dynamics studies[35] have indicated that the central rotor can be flexible, the present work provides a better understanding of how this flexibility is both conferred and modulated by the εCTD. When the εCTD is in the up or half-up conformation, the εCTH1 binds to the γ subunit, clamping the central rotor together (Figs. 8 and 9). This clamping would stabilize the central stalk in a closed position, increasing the stiffness of the rotor which, in turn, has the potential to impede enzyme turnover by decreasing the flexibility of the intact enzyme. When the εCTD subunit is in the down conformation, εCTH1 can no longer bridge between the εNTD and the γ subunit, thereby enabling the complex to open up to an unclamped conformation and allow a greater degree of flexible coupling between the $F_1$ and $F_o$ motors (Figs. 6, 8 and 9). The ATPase assays presented here suggest that, when the enzyme is in half-up conformation (εΔCTH2), although the turnover is reduced compared to a fully active enzyme (εΔCTH1 + 2), it remains greater than for an enzyme that is able to be fully autoinhibited (WT) (Fig. 7). Together these data indicate that the εCTD is able to modulate central rotor flexibility in addition to inserting into the $F_1$-ATPase to physically block rotation. An in-depth study on yeast $F_1F_o$ ATP synthase[46], using similar methods to those in this study and which presented the enzyme after incubation with 10 mM ATP, revealed the enzyme in many rotary states. In all structures the rotor was in the same conformation and showed none of the torsional flexing that we observed in the present study, suggesting that the torsional flexing we observe may be limited to the bacterial or *E. coli* enzyme, and may not always be seen with other species.

The precise function of ATP in regulating the ε subunit of $F_1F_o$ ATP synthase appears to vary between bacterial species. For example, the εCTD of *Bacillus* PS3 $F_1F_o$ ATP synthase forms a single helix[49] that does not bridge the γ subunit in the same manner as the *E. coli* εCTD (Supplementary Fig. 9). Furthermore, the isolated *Bacillus* PS3 ε subunit has been shown to bind ATP[50–52] with micromolar binding affinity ($Kd = 4.3\,\mu M$[53]) and a FRET sensor based on this subunit has been used to visualize ATP levels inside living cells[54]. Hence, in *Bacillus* PS3, cellular

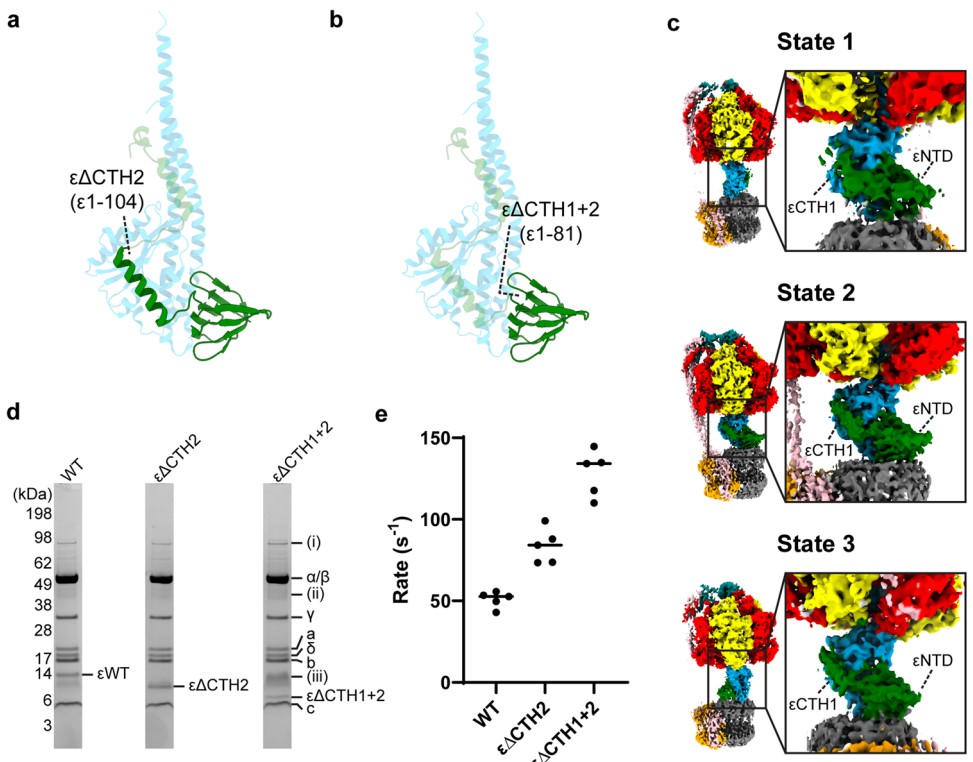

**Fig. 7 Subunit ε attenuates ATPase activity. a** εΔCTH2 truncation retains the εCTH1 which can bind to the γ subunit. **b** εΔCTH1 + 2 retains only the εNTD akin to the εCTD down conformation. **c** Cryo-EM maps of the 3 states generated using the εΔCTH2 truncation mutant after exposure to 10 mM MgATP showed *E. coli* F₁Fₒ ATP synthase in the half-up sub-state, with εCTH1 attached to the subunit γ. The 3 states have been rotated successively by 120° to show the position of subunit ε. **d** Coomassie stained SDS PAGE (uncropped image provided as Supplementary Data 5) of purified *E. coli* F₁Fₒ ATP synthase WT, εΔCTH2 and εΔCTH1 + 2. Subunits labeled and minor contamination bands identified as; (i) Ribonuclease E, (ii) GroEL, and (iii) ElaB. **e** ATP regeneration assays of WT (containing full length subunit ε), εΔCTH2 (ε1–104) and εΔCTH1 + 2 (ε1–81). All data points and mean are shown (raw traces are in Supplementary Fig. 8 and values in Supplementary Data 6). Removal of εCTH2 results in higher ATP turnover than WT. Removal of both εCTH1 and εCTH2 shows higher ATP turnover than removal of only εCTH2.

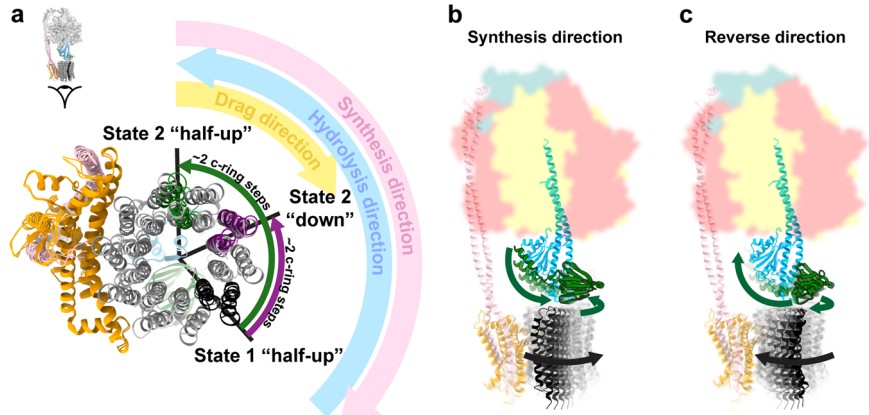

**Fig. 8 Rotation in the Fₒ motor in the half-up and down sub-states. a** The Fₒ rotor of the State 2 down sub-state trails the State 2 half-up sub-state by two c subunit steps in the synthesis direction. **b** When the Fₒ motor is driven in the synthesis direction, the central stalk is unclamped and the εNTD is pushed away from the εCTH1 binding site on γ, reducing the likelihood of the up conformation. **c** When the Fₒ motor is driven in the reverse direction, the central stalk is closed and the εNTD is pushed towards the εCTH1 binding site on γ, increasing the likelihood of the up conformation.

ATP concentrations are likely to regulate the εCTD conformation and the single εCTD helix is unable to clamp the central stalk in the manner we observe in *E. coli* ATP synthase. The ATP binding affinity for the isolated *E. coli* subunit ε has been shown to be much weaker than *Bacillus* PS3, with little discrimination between ATP and ADP (*Kd* of ~20 mM for both ATP and ADP[13,52]). The crystal structure of the isolated ε subunit from *E. coli* also demonstrates that the εCTD can form the down conformation in the absence of ATP[20]. Further to this, studies on *Caldalakliba-cillus thermarum* F₁-ATPase have also shown that removal of the ε subunit ATP binding site has little effect on activity or conformation of the εCTD[55]. Hence, other mechanisms in addition to ATP binding could potentially control the conformation of the εCTD in *E. coli* and other related bacteria.

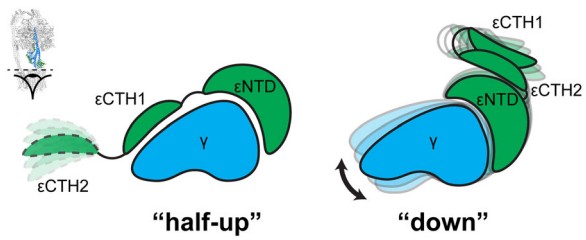

**Fig. 9 Schematic of clamping by εCTH1.** When the εCTH1 is in the half-up state and bound to subunit γ, the central stalk is clamped and increases the stiffness of the rotor. When εCTH1 is in the down state, the central stalk is no longer clamped and has increased flexibility.

The cryo-EM maps generated in this study suggest another mechanism whereby the conformation of subunit ε could be controlled by the pmf. When $F_1F_o$ ATP synthase is operating in synthesis mode with high pmf, torque generated in the $F_o$ motor would drive the $F_o$ ring clockwise (Fig. 8b). This torque would pull the εNTD clockwise and increase the distance between it and the εCTH1 binding site on subunit γ, reducing the likelihood of εCTH1 binding and favoring the εCTD down conformation (Figs. 6 and 8b). If the pmf were to be reversed, the torque in $F_o$ would rotate the εNTD anticlockwise and decrease the distance to the εCTH1 binding site on subunit γ, increasing the likelihood of εCTH1 binding and consequently the εCTD up conformation (Figs. 6 and 8c). In this way, the function of *E. coli* $F_1F_o$ ATP synthase could be modulated by a combination of the cellular ATP concentration and the pmf (i.e., the torque in the $F_o$ motor). Hence, the εCTD would act as a ratchet, inhibiting $F_1F_o$ ATP synthase when the pmf is insufficient to drive ATP synthesis, allowing the bacterium to grow and quickly adapt to unfavorable conditions. A recent in vivo crosslinking study investigating the physiological relevance of subunit ε demonstrated that the εCTD down state would be most prevalent in conditions that produce high pmf[56], suggesting that the ε subunit does change conformation in response to pmf in addition to ATP concentration. Hence, the data presented here illustrates both flexible coupling in the central stalk and a potential mechanism whereby this torsional flexing could impact the regulation of $F_1F_o$ ATP synthase.

## Methods
**Cloning and protein purification.** Mutant constructs were made by overlap extension PCR and using the following primers:

εΔCTH2：Forward primer 5'-aagcgaaacgtaaggctgaagagcactaacaccggcttgaaaagca caaa-3'

Reverse primer 5'-tggcttttgtgctttttcaagccggtgttagtgctcttcagccttacgttt-3'

εΔCTH1 + 2: Forward primer 5'-aacgtgaccgttctggccgactaacaccggcttgaaaagcaca aa-3'

Reverse primer 5'-ggctttttgtgctttttcaagccggtgttagtcggccagaacggtcacgtt-3'

*E. coli* $F_1F_o$ ATP synthase protein was prepared as described in Sobti et al.[21,25]. Cysteine-free *E. coli* ATP synthase (all cysteine residues substituted with alanine and a His-tag introduced on the β subunit) was expressed in *E. coli* DK8 strain[22]. Cells were grown at 37 °C in LB medium supplemented with 100 μg/ml ampicillin for 5 h. The cells were harvested by centrifugation at 5000 × g, providing ~1.25 g cells per liter of culture. Cells were resuspended in lysis buffer containing 50 mM Tris/Cl pH 8.0, 100 mM NaCl, 5 mM MgCl₂, 0.1 mM EDTA, 2.5% glycerol and 1 μg/ml DNase I, and processed with three freeze thaw cycles followed by one pass through a continuous flow cell disruptor at 20 kPSI. Cellular debris was removed by centrifuging at 7700 × g for 15 min, and the membranes collected by ultracentrifugation at 100,000 × g for 1 h. The ATP synthase complex was extracted from membranes at 4 °C for 1 h by resuspending the pellet in extraction buffer consisting of 20 mM Tris/Cl, pH 8.0, 300 mM NaCl, 2 mM MgCl₂, 100 mM sucrose, 20 mM imidazole, 10% glycerol, 4 mM digitonin and EDTA-free protease inhibitor tablets (Roche). Insoluble material was removed by ultracentrifugation at 100,000 × g for 30 min. The complex was then purified by binding on Talon resin (Clontech) and eluted in 150 mM imidazole, and further purified with size exclusion chromatography on a 16/60 Superose 6 column equilibrated in a buffer containing 20 mM Tris/Cl pH 8.0, 100 mM NaCl, 1 mM digitonin, and 2 mM MgCl₂. The purified WT protein was then concentrated to 11 μM (6 mg/ml), and

snap frozen and stored for grid preparation while the εΔCTH2 and εΔCTH1 + 2 mutants were concentrated and frozen at 9 μM (5 mg/ml)

**Cryo-EM grid preparation.** One μl of 100 mM ATP/100 mM MgCl₂ (pH 8.0) was added to an aliquot of 9 μl of purified cysteine-free *E. coli* $F_1F_o$ ATP synthase (WT at 11 μM and εΔCTH2 mutant at 9 μM) and the sample was incubated at 22 °C for 30 s, before 3.5 μl was placed on glow-discharged holey gold grid (UltrAufoils R1.2/1.3, 200 Mesh). Grids were blotted for 4 s at 22 °C, 100% humidity and flash-frozen in liquid ethane using a FEI Vitrobot Mark IV (total time for sample application, blotting, and freezing was 45 s).

**WT cryo-EM data collection and data processing.** Grids were transferred to a Thermo Fisher Talos Arctica transmission electron microscope (TEM) operating at 200 kV and screened for ice thickness and particle density. Suitable grids were subsequently transferred to a Thermo Fisher Titan Krios TEM operating at 300 kV equipped with a Gatan BioQuantum energy filter and K3 Camera at the Pacific Northwest Centre for Cryo-EM at OHSU. Images were recorded automatically using SerialEM v3.7 at ×81,000 magnification yielding a pixel size of 0.54 Å (K3 operating in super resolution mode). A total dose of 48 electrons per Å² was used spread over 77 frames, with a total exposure time of 3.5 s. In all, 8620 movie micrographs were collected (Supplementary Fig. 1). MotionCorr2[57] was used to correct local beam-induced motion and to align resulting frames, with 9 × 9 patches and binning by a factor of two. Defocus and astigmatism values were estimated using Gctf[37] and 8215 micrographs were selected after exclusion based on ice contamination, drift, astigmatism. Approximately 1000 particles were manually picked and subjected to 2D classification to generate templates for template picking in cryoSPARC[58], yielding 869,147 particles. These particles were binned by a factor of four and subjected to 2D classification generating a final dataset of 429,638 particles. The locations of these particles were then imported into Relion[59], re-extracted at full resolution, and further classified into 3D classes using a low pass filtered cryo-EM model generated from a previous study[18], yielding the three main states related by a rotation of the central stalk (State 1, State 2, and State 3 with 100,831, 215,003, and 113,804 particles, respectively). Focused classification, using a mask comprising the lower half of the central rotor, was implemented without performing image alignment in Relion 3.0, yielding the half-up and the down sub-classes in each of the three main states. A further $F_o$ focused classification without image shifts was performed on each of the half-up and down sub-classes to elucidate the position of $F_o$ subunits in the respective sub-states. See Supplementary Fig. 2 for a flowchart describing this classification.

**εΔCTH2 mutant cryo-EM data collection and data processing.** Grids were transferred to a Thermo Fisher Talos Arctica transmission electron microscope (TEM) operating at 200 kV and screened for ice thickness and particle density. Suitable grids were subsequently transferred to a Thermo Fisher Titan Krios TEM operating at 300 kV equipped with a Gatan BioQuantum energy filter and K2 Camera at the Molecular Horizons, University of Wollongong. Images were recorded automatically using EPUv2.7 at ×120,000 magnification yielding a pixel size of 1.13 Å. A total dose of 55 electrons per Å² was used spread over 50 frames, 3680 movie micrographs were collected. All the processing was subsequently performed in cryoSPARC[58]. Initial particles were picked using the blob protocol which were 2D classified to create templates to the pick the entire dataset. Extracted particles were subjected to multiple rounds of 2D classification, ab initio reconstruction, heterogenous refinement to sort the particles into discreet structures. See Supplementary Fig. 7 for a flowchart describing this classification. Additional processing was performed using masks and focused refinement but did not yield any maps showing alternate conformations, so are not included in the flow chart. Supplementary Table 1 contains a summary of data collection/processing statistics and Supplementary Fig. 10 for FSC curves.

**Model building.** Models were built and refined in Coot[60], PHENIX[29], and ISOLDE[61] using pdbs 6OQT, 6OQV, 6OQW[19] (*E. coli* ATP synthase incubated with MgADP), and 1AQT[20] (isolated *E. coli* ATP synthase subunit ε) as guides. See Supplementary Table 1 for a summary of refinement and validation statistics.

**ATP regeneration assays.** ATP regeneration assays were performed as in Sobti et al.[25]. Five μg of detergent solubilized protein was added to 100 mM KCl, 50 mM MOPS pH 7.4, 1 mM MgCl₂, 1 mM ATP, 2 mM PEP, 2.5 units/ml pyruvate kinase, 2.5 units/ml lactate dehydrogenase and 0.2 mM NADH and monitored for OD at 340 nm at for up to 20 min (Supplementary Fig. 8).

**Statistics and reproducibility.** The cryo-EM analyses were performed on single protein preparations. We have performed similar experiments on similar protein preparations (more than ten times using a range of microscopes) with similar outcomes[18,19,21,25]. Detailed statistics for the sample size, data collection and analysis of cryo-EM data are provided in Supplementary Table 1, and Fourier shell correlation curves are provided in Supplementary Fig. 10. The activity assays were performed in technical triplicate, with only means provided as the raw data

(Supplementary Fig. 8 and Data 6) could be plotted and interpreted without error bars (Fig. 7e).

**Reporting summary**. Further information on research design is available in the Nature Portfolio Reporting Summary linked to this article.

## Data availability
The models generated and analyzed during the current study are available from the RCSB PDB: 8DBP, 8DBQ, 8DBR, 8DBS, 8DBT, 8DBU, 8DBV, 8DBW. The cryo-EM maps used to generate models are available from the EMDB: 27296, 27297, 27298, 27299, 27300, 27301, 27302, 27303, 27304, 27305, 27306, 27307, 27308, 27309, 27310, 27311, 27312, 27313, 27314, 27315. Phenix combined maps and models for State 2 up and down are provided as Supplementary Data 1–4 with this manuscript.

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

## Acknowledgements
We wish to thank and acknowledge Dr. Craig Yoshioka and Dr. Claudia López (Oregon Health & Sciences University (OHSU)) for data collection and processing expertise. We would also like to thank Dr Emily Furlong (The Victor Chang Cardiac Research Institute) for critically reviewing the manuscript. A.G.S. was supported by a National Health and Medical Research Council Fellowship APP1159347 and Grant APP1146403. We acknowledge the use of the Victor Chang Innovation Centre, funded by the NSW Government, and the Electron Microscope Unit at UNSW Sydney, funded in part by the NSW Government. We also acknowledge he use of the University of Wollongong Cryogenic Electron Microscopy Facility at Molecular Horizons. A portion of this research was supported by NIH grant U24GM129547 and performed at the Pacific Northwest Centre for Cryo-EM at OHSU and accessed through EMSL (grid.436923.9), a DOE Office of Science User Facility sponsored by the Office of Biological and Environmental Research.

## Author contributions
M.S., R.I., and A.G.S. conceived the study and wrote the manuscript. M.S. performed the formal analysis of the study. Y.C.Z. performed the reconstitution for ATP regeneration assays and edited the manuscript. J.L.W. and S.H.J.B. aided in cryo-EM data acquisition and analysis. A.G.S. supervised the study.

## Competing interests
The authors declare no competing interests.
