## [Peer Review File · Communications Biology]

Reviewers' comments:

Reviewer #1 (Remarks to the Author):

The MS by Sobti et al. describes different states of E. coli ATP synthase in presence of 10 mM MgATP, as observed by cryo-EM. The study is well designed and carefully executed, and presents interesting novel results, shedding some light on the role of the epsilon subunit during hydrolysis. Still, there are a few questions remaining open.

The co-existence of two conformations of the C-terminal domain of subunit epsilon, "half-up" and "down" is puzzling. What is the physiological importance of each state? Does the enzyme go through both conformations for each hydrolyzed ATP? This does not seem likely as getting into the "half-up" conformation is associated with a 2 c subunit rotation in synthesis direction. What determines if a specific enzyme molecule has a "half-up" or a "down" conformation?

In the present study, removal of the C-terminal domain of epsilon increases the activity by about 150%. In contrast, in isolated F1 removal of epsilon increases the activity ten-fold (Dunn, 1982; Weber et al., 1998). Does this mean that the remaining N-terminal domain of epsilon still inhibits?

Reviewer #2 (Remarks to the Author):

The authors present cryo-EM structures of the E. coli F1FO ATP synthase in the presence of 10 mM ATP that resolve new conformations of this rotary motor that is responsible for the synthesis of the majority of cellular ATP. These structures have the potential to provide new insight into the rotary mechanism of this molecular motor. As such, the results are of sufficient consequence to be considered for publication in Nature Communications.

However, there are concerns that must be adequately addressed before a decision can be made regarding whether the manuscript is acceptable for publication.

1. Lines 46-47: "However, cells have evolved inhibitory mechanisms to avoid wasteful hydrolysis of ATP that could occur under many physiological conditions."

Line 54: "The ϵ CTD can inhibit the enzyme by inserting into the F1 motor and physically blocking rotation."

Lines 71-74: "We have used cryo-EM to examine detergent-solubilized E. coli F1Fo ATP synthase²⁰ following a 45 second incubation with 10 mM MgATP. These conditions allow the enzyme to be observed operating in the hydrolysis direction, and are similar to the concentrations of ATP and ADP found in E. coli undergoing aerobic respiration²¹."

There are problems with these statements that must be addressed. Lines 46-47 and 54 appear to contradict Lines 71-74. Because these statements provide the premise for the study, it is important that this potential contradiction be addressed.

The claim that these conditions were similar to the concentrations of ATP and ADP found in E. coli undergoing aerobic respiration is incorrect because ADP and Pi were absent during the incubation. To determine the free energy of the chemical potential of ATP in the cell, the concentrations of ATP, ADP, and Pi must be measured. For E. coli, these were first measured by Kashet (1982) Biochemistry 21: 5534-5538 who determined these values when E. coli were growing at three different pH values and got 3.3 mM ATP, 0.4 mM ADP and 5.6 mM Pi at pH 7.25. Reference 21 reports what appears to be more quantitative measurements of 9.6 mM ATP and 0.56 mM ADP, but fails to report Pi. Using 5.6 mM Pi, the free energy values of ATP hydrolysis differ by 1.5 fold (11.7 kcal/mol for Kashket versus 15.34 kcal/mol for Ref.21).

However, the conditions for cryo-EM in the manuscript did not include ADP and Pi. Consequently, the free energy of ATP hydrolysis was much higher than 15.34 kcal/mol. With regard to Lines 46-47 and 54, the authors must explain why their results (Fig 3) showing that using 10 mM ATP without ADP and Pi promotes the ATPase-activated ϵ CTD conformation that is withdrawn from F1

motor.

2. The authors clearly show in Fig 7 that F₁F_o is actively hydrolyzing ATP under the conditions in which the cryo-EM structures were determined. If the motors were rotating continuously up to the point at which they were frozen, it is anticipated that this would result in many structures with the rotor in a variety of rotary positions from 0 to 120 degrees. However, this was not observed. Instead, the rotary positions of each of the three States varied over a narrow range. Consequently, the conditions caused the motor to spend most of its time in a few stable low free energy states relative to other possible rotary positions even in the ATPase-activated, εCTD withdrawn conformation. The authors need to address a possible reason for this disparity in the Discussion.

3. The authors also need to provide information regarding the rotary position of subunit-γ in this manuscript relative to the rotary positions during the F₁-ATPase catalytic cycle.

Lines 94-95: "Compared to the same enzyme imaged in the presence of 10 mM MgADP18, the central stalk had rotated as a rigid body by ~10 degrees."

The authors need to state whether this is CW or CCW relative to the 10 mM MgADP condition. They must also provide a context of the rotary position relative to F₁ structures including PDB structures 2JDI, the ground state structure (catalytic dwell), 4ASU, 3OAA, and 7L1Q as discussed in Sobti et al. (2021) Nat Comm 12:4690 and in Frasch et al. (2022) Front. Microb. Fmicrob 2022.965620.

4. Lines 142-143: "The sub-states of State 2 showed the best detail for the two εCTD conformations and allowed the c-subunits to be assigned to F_o rotational sub-states based on their interaction with the εNTD..."

Lines 148-153: "When the State-2 half-up and down structures were compared additional clear differences beyond the εCTD up and down conformations were observed. When aligned on the stator a subunit, the F_o ring rotates the equivalent of two c-subunits in a clockwise direction when viewed from the membrane (akin to the synthesis direction). This rotation of the F_o motor was facilitated mainly by a twisting of the central stalk (~50degrees) but also by flexing of the peripheral stalk (~15 degrees)..."

The authors need to discuss their structure that shows rotation of the F_o ring relative to that reported via single-molecule rotation studies by Yanagisawa and Frasch ((2017) JBC 292: 17093-17100 and (2021) eLife 10:e70016). These appear to be similar effects that occur under similar conditions at similar rotary positions. The F_o ring rotation structure reported by the authors was most clearly resolved in State 2. Sielaff et al. ((2019) Molecules 24:504-529) reported similar asymmetric effects of F₁F_o by a variety of single-molecule approaches, and correlated the asymmetries to the three F₁F_o states observed by cryo-EM. The authors need to address whether their current results are consistent with the conclusions of Sielaff et al.

5. Lines 231-233: "Although flexible coupling between F₁ and F_o motors is necessary to facilitate efficient enzyme function, whether this flexibility originates from the peripheral or central stalk has been controversial (refs28-33)." Lines 248-249: "Although single molecule (refs29,39) and molecular dynamics studies (ref28) have indicated that the central rotor can be flexible,..."

References 28 and 29 report elasticity of F₁. The authors need to reference Martin et al. (2018) PNAS 115: 5750-5755, which showed direct evidence of elasticity with spring constants comparable to those of Ref 28, which occurred over similar rotary positions as that observed by the structures presented in the current manuscript. Lines 245-246: "We propose that the "drag" that would be experienced at the stator/lipid/detergent interfaces results in the "delay" of the F_o ring during rotation that we observe here (Fig 8a)."

The authors need to discuss an alternative basis for the "drag" that they observe, which results from the interaction between the c-ring and subunit-a as reported by Yanagisawa and Frasch (2021) eLife 10:e70016. This conditions in which this latter interaction was observed are closely similar to those reported here, and both occur over apparently similar rotary positions of the motor.

6. Minor comment: in Fig 4, the difference in color between the grey and black c-subunits is currently insufficient to distinguish the black one.

Reviewer #3 (Remarks to the Author):

The studies in this manuscript investigate the structural changes in the E. coli ATP synthase in the transition from the inhibited state to the active state with the addition of Mg:ATP. A key finding is that the transition to the active state is associated with considerable torsional flexing of the central stalk. An impetus of this study is to understand how the enzyme manages the torsional stress that is a result of the mismatch between the number of protons, 10, that are used for the synthesis of 3 ATPs and the 3 fold symmetry of the enzyme. Both the central and peripheral stalks have been suggested to store torsional energy during the reaction cycle.

This study uses cryo-EM to study a series of intermediate structures after the addition of Mg:ATP. The study also includes a couple of variants in the epsilon subunit. The studies are well done and the structures are good. The conclusions are consistent with the results and provide structural evidence that the central stalk twists in the transition from the inactive state to the active state. There are also observed changes in the position of the peripheral stalk. Overall, this is a good study and does provide some new information.

The only recommendation is that these results are discussed in reference of the paper by Guo and Rubinstein (Structure of ATP synthase under strain during catalysis. Guo H, Rubinstein JL. Nat Commun. 2022 Apr 25;13(1):2232.). This study studies the yeast ATP synthase with similar goals and methods, but come to some different conclusions. Specifically, they conclude that the central stalk does not distort during ATP hydrolysis and they have a very impressive collection of structures that seem to cover the entire reaction pathway.

Reviewer #1 (Remarks to the Author):

The MS by Sobti et al. describes different states of E. coli ATP synthase in presence of 10 mM MgATP, as observed by cryo-EM. The study is well designed and carefully executed, and presents interesting novel results, shedding some light on the role of the epsilon subunit during hydrolysis. Still, there are a few questions remaining open.

We thank the reviewer for their helpful and constructive comments, and have replied to these below.

The co-existence of two conformations of the C-terminal domain of subunit epsilon, “half-up” and “down” is puzzling. What is the physiological importance of each state? Does the enzyme go through both conformations for each hydrolyzed ATP? This does not seem likely as getting into the “half-up” conformation is associated with a 2 c subunit rotation in synthesis direction. What determines if a specific enzyme molecule has a “half-up” or a “down” conformation?

Our study provides no evidence on whether the up/down transition occurs for each hydrolyzed ATP, as we rely on averaging methods to generate the maps and cannot track individual molecules during rotation. Our hypothesis is that rotation of the c-ring in the ATP synthesis direction favors the down state, as it this would pull the epsilon N-terminal domain away from the gamma binding site (Figure 8). We have modified the discussion to improve clarity.

In the present study, removal of the C-terminal domain of epsilon increases the activity by about 150%. In contrast, in isolated F1 removal of epsilon increases the activity ten-fold (Dunn, 1982; Weber et al., 1998). Does this mean that the remaining N-terminal domain of epsilon still inhibits?

We can provide no evidence to why our studies show a difference in this regard. One possibility is that we have purified the intact protein via size exclusion chromatography using the detergent digitonin (whereas Dunn, 1982 and Weber et al., 1998 performed studies on the isolated F1 enzyme without Fo). All other detergents we trialed resulted in the enzyme disassociating into the F1 and Fo subcomplexes (particularly over size exclusion but also when imaged using electron microscopy). In this study the activity is lower and a possibility for this may be drag encountered in the Fo motor (as discussed in the submitted manuscript). Hence rather than the N-terminal domain of epsilon inhibiting, it could be the detergent solubilized Fo motor reducing turnover.

Reviewer #2 (Remarks to the Author):

The authors present cryo-EM structures of the E. coli F1FO ATP synthase in the presence of 10 mM ATP that resolve new conformations of this rotary motor that is responsible for the synthesis of the majority of cellular ATP. These structures have the potential to provide new insight into the rotary mechanism of this molecular motor. As such, the results are of sufficient consequence to be considered for publication in Nature Communications.

However, there are concerns that must be adequately addressed before a decision can be

made regarding whether the manuscript is acceptable for publication.

Again, we thank the reviewer for their helpful and constructive comments, and have replied to these below.

1. Lines 46-47: “However, cells have evolved inhibitory mechanisms to avoid wasteful hydrolysis of ATP that could occur under many physiological conditions.”

Line 54: “The ϵ CTD can inhibit the enzyme by inserting into the F1 motor and physically blocking rotation.”

Lines 71-74: “We have used cryo-EM to examine detergent-solubilized *E. coli* F1Fo ATP synthase²⁰ following a 45 second incubation with 10 mM MgATP. These conditions allow the enzyme to be observed operating in the hydrolysis direction, and are similar to the concentrations of ATP and ADP found in *E. coli* undergoing aerobic respiration²¹.”

There are problems with these statements that must be addressed. Lines 46-47 and 54 appear to contradict Lines 71-74. Because these statements provide the premise for the study, it is important that this potential contradiction be addressed.

The claim that these conditions were similar to the concentrations of ATP and ADP found in *E. coli* undergoing aerobic respiration is incorrect because ADP and Pi were absent during the incubation. To determine the free energy of the chemical potential of ATP in the cell, the concentrations of ATP, ADP, and Pi must be measured. For *E. coli*, these were first measured by Kashet (1982) *Biochemistry* 21: 5534-5538 who determined these values when *E. coli* were growing at three different pH values and got 3.3 mM ATP, 0.4 mM ADP and 5.6 mM Pi at pH 7.25. Reference 21 reports what appears to be more quantitative measurements of 9.6 mM ATP and 0.56 mM ADP, but fails to report Pi. Using 5.6 mM Pi, the free energy values of ATP hydrolysis differ by 1.5 fold (11.7 kcal/mol for Kashket versus 15.34 kcal/mol for Ref.21).

However, the conditions for cryo-EM in the manuscript did not include ADP and Pi. Consequently, the free energy of ATP hydrolysis was much higher than 15.34 kcal/mol. With regard to Lines 46-47 and 54, the authors must explain why their results (Fig 3) showing that using 10 mM ATP without ADP and Pi promotes the ATPase-activated ϵ CTD conformation that is withdrawn from F1 motor.

As shown in our previous study (Sobti et al. 2019) ~0.25 mM ADP and Pi should be present in this system at the point of freezing. It is common to perform activity assays in the absence of Pi. Purification of this enzyme in high concentration phosphate buffers impeded structure solution via cryoEM (attempted for previous studies) and therefore Tris buffer (without Pi) was used. Although we are unable to exactly recreate the exact conditions observed during *E. coli* aerobic respiration, the conditions we use here are comparable to that observed in cells and near identical methods (addition of 10 mM ATP to protein in Tris buffer) have now been performed by others on a related enzyme (Guo & Rubinstein 2022). The conditions are not identical to those found in cells, but are a close approximation.

Moreover, single molecule studies on lipid nanodisc solubilized *E. coli* F1Fo enzyme (DOI: 10.1038/emboj.2010.259) use similar conditions as in submitted manuscript, with the assay buffer containing 10 mM KCl, 50 mM Tris, pH 8.0, PEG400, 2 mM ATP and 1 mM MgCl₂.

We have edited the following lines:

Lines 46-47: “However, cells have evolved inhibitory mechanisms to avoid wasteful hydrolysis of ATP that could occur under **certain** physiological conditions.”

Lines 54-55: “**It has been hypothesized that the ϵ CTD is able to inhibit the F₁ motor by extending “up” and blocking rotation under certain conditions^{16,17}**”

Lines 71-74: “These conditions allow the enzyme to be observed operating in the hydrolysis direction, and are similar to the concentrations found in *E. coli* undergoing aerobic respiration²¹ (**though with much lower Pi concentration²²**).”

2. The authors clearly show in Fig 7 that F₁F_o is actively hydrolyzing ATP under the conditions in which the cryo-EM structures were determined. If the motors were rotating continuously up to the point at which they were frozen, it is anticipated that this would result in many structures with the rotor in a variety of rotary positions from 0 to 120 degrees. However, this was not observed. Instead, the rotary positions of each of the three States varied over a narrow range. Consequently, the conditions caused the motor to spend most of its time in a few stable low free energy states relative to other possible rotary positions even in the ATPase-activated, ϵ CTD withdrawn conformation. The authors need to address a possible reason for this disparity in the Discussion.

It is likely that we only observe the molecule in three rotary states as the enzyme spends the majority of the time in these low energy states. The lipid nanodisc solubilized enzyme has been observed by single molecule methods (DOI: 10.1038/emboj.2010.259) and without external load on the enzyme (in the form of increased medium viscosity), *E. coli* F₁F_o showed little to no sub stepping. Assuming it behaves similar to the isolated F₁ ATPase, it would have an 8 ms dwell time and a 0.27 ms rotation time, indicating that *E. coli* F₁F_o would spend ~97 % of the time in catalytic dwell. Hence it is unlikely that we would be able to observe the enzyme in sub stepping under these conditions.

We have expanded the discussion to explore this:

“Although the enzyme is likely rotating and hydrolyzing ATP during the freezing process²⁴, we did not observe sub steps (e.g. the binding dwell) in the enzyme beyond the three catalytic dwells and epsilon/c-ring sub rotation. This is likely due to the limited time the enzyme would spend outside the catalytic dwell under these imaging conditions, with single molecule studies needing external load, in the form of increased medium, to observe substates⁹. Assuming that the F₁F_o enzyme has similar turnover to the F₁-ATPase, single molecule studies suggest that the enzyme would be in the catalytic dwell ~97% of the time³³, and hence too small number of particles for efficient sorting.”

3. The authors also need to provide information regarding the rotary position of subunit- γ in this manuscript relative to the rotary positions during the F₁-ATPase catalytic cycle.

Lines 94-95: “Compared to the same enzyme imaged in the presence of 10 mM MgADP18, the central stalk had rotated as a rigid body by ~10 degrees.”

The authors need to state whether this is CW or CCW relative to the 10 mM MgADP condition. They must also provide a context of the rotary position relative to F₁ structures including PDB structures 2JDI, the ground state structure (catalytic dwell), 4ASU, 3OAA, and 7L1Q as discussed in Sobti et al. (2021) Nat Comm 12:4690 and in Frascch et al. (2022)

Front. Microb. Fmicrob 2022.965620.

We agree with the reviewer that this should have been clearer in the submitted manuscript.

We have now calculated the rotation angle of the axel of each structure suggested by the reviewer relative to the bMF1 ground state (bacteria residues 1-22 & 247-284; bMF1 1-22 & 233-273) using Chimera.

The rotation angles counterclockwise when viewed from membrane (hydrolysis direction) are as follows:

PS3 F1-ATPase catalytic dwell (7L1R): +5°

E. coli +10 mM MgATP (state 2 from this study): +5°

E. coli +10 mM MgATP (6OQV): +14°

E. coli F1-ATPase epsilon inhibited (3OAA): +18°

Bovine F1-ATPase all sites nucleotide bound (4ASU): +18°

PS3 F1-ATPase binding dwell (7L1Q): +53°

We have expanded lines 94-95 to describe the relative rotation angles:

“Compared to the same enzyme imaged in the presence of 10 mM MgADP¹⁹, the central stalk had rotated as a rigid body by ~10° counterclockwise, when viewed from the membrane (Fig. 2b and Extended Data Fig. 3), the εCTH2 has dissociated from the central stalk, and the β1 (β_{DP}) subunit has closed to contact the γ subunit (Fig. 2c) and bind MgADP. Comparison of the relative position of the rotor axel between known structures^{8,16,19,26-28}, highlighted that the F1-ATPase was in a similar rotary position to that of observed of *Geobacillus stearothermophilus* (also termed *Bacillus* PS3) in the catalytic dwell⁸ (Extended Data Fig. 3), indicating that this enzyme was in a similar state.”

4. Lines 142-143: “The sub-states of State 2 showed the best detail for the two εCTD conformations and allowed the c-subunits to be assigned to Fo rotational sub-states based on their interaction with the εNTD...”

Lines 148-153: “When the State-2 half-up and down structures were compared additional clear differences beyond the eCTD up and down conformations were observed. When aligned on the stator a subunit, the Fo ring rotates the equivalent of two c-subunits in a clockwise direction when viewed from the membrane (akin to the synthesis direction). This rotation of the Fo motor was facilitated mainly by a twisting of the central stalk (~50degrees) but also by flexing of the peripheral stalk (~15 degrees)...”

The authors need to discuss their structure that shows rotation of the Fo ring relative to that reported via single-molecule rotation studies by Yanagisawa and Frasch ((2017) JBC 292: 17093-17100 and (2021) eLife 10:e70016). These appear to be similar effects that occur under similar conditions at similar rotary positions. The Fo ring rotation structure reported by the authors was most clearly resolved in State 2. Sielaff et al. ((2019) Molecules 24:504-529) reported similar asymmetric effects of F1Fo by a variety of single-molecule approaches, and correlated the asymmetries to the three F1Fo states observed by cryo-EM. The authors need to address whether their current results are consistent with the conclusions of Sielaff et al.

We thank the reviewer for bring these studies to our attention. We have now compared the Fo ring rotation we observed in cryo-EM to other studies and included this in the revised manuscript:

“Moreover, the relative number of particles between each state differed substantially and the proportional differences were different than that observed for the same enzyme incubated with MgADP¹⁹. Other work using single molecule methods has suggested that the 3:10 symmetry mismatch between the F₁ and F_o motors would cause asymmetry in the c-ring rotation³³⁻³⁵, and the structural data of *E. coli* F₁F_o incubated with either MgADP¹⁹ or MgATP certainly corroborates this.”

5. Lines 231-233: “Although flexible coupling between F₁ and F_o motors is necessary to facilitate efficient enzyme function, whether this flexibility originates from the peripheral or central stalk has been controversial (refs28-33).” Lines 248-249: “Although single molecule (refs29,39) and molecular dynamics studies (ref28) have indicated that the central rotor can be flexible,...”

References 28 and 29 report elasticity of F₁. The authors need to reference Martin et al. (2018) PNAS 115: 5750-5755, which showed the stator/lipid/detergent interfaces results in the “delay” of the F_o ring during rotation that direct evidence of elasticity with spring constants comparable to those of Ref 28, which occurred over similar rotary positions as that observed by the structures presented in the current manuscript. Lines 245-246: “We propose that the “drag” that would be experienced at we observe here (Fig 8a).”

The authors need to discuss an alternative basis for the “drag” that they observe, which results from the interaction between the c-ring and subunit-a as reported by Yanagisawa and Frasch (2021) eLife 10:e70016. This conditions in which this latter interaction was observed are closely similar to those reported here, and both occur over apparently similar rotary positions of the motor.

Martin et al. (2018) PNAS 115: 5750-5755 is centered on the F₁-ATPase, not drag at the stator interface.

We have now included discussion of these works in the Discussion:

“The counter rotation we observe here could be due to several factors. Single molecule studies^{34,47} have identified rotation in F_o motor of a similar magnitude and direction and therefore, suggesting this could be indicative of sub stepping in the synthesis direction.”

6. Minor comment: in Fig 4, the difference in color between the grey and black c-subunits is currently insufficient to distinguish the black one.

We have added an arrow to clearly label the c-subunit.

Reviewer #3 (Remarks to the Author):

The studies in this manuscript investigate the structural changes in the *E. coli* ATP synthase in the transition from the inhibited state to the active state with the addition of Mg:ATP. A key finding is that the transition to the active state is associated with considerable torsional flexing of the central stalk. An impetus of this study is to understand how the enzyme manages the torsional stress that is a result of the mismatch between the number of protons, 10, that are used for the synthesis of 3 ATPs and the 3 fold symmetry of the enzyme. Both the central and peripheral stalks have been suggested to store torsional energy during the reaction cycle.

This study uses cryo-EM to study a series of intermediate structures after the addition of Mg:ATP. The study also includes a couple of variants in the epsilon subunit. The studies are well done and the structures are good. The conclusions are consistent with the results and provide structural evidence that the central stalk twists in the transition from the inactive state to the active state. There are also observed changes in the position of the peripheral stalk. Overall, this is a good study and does provide some new information.

The only recommendation is that these results are discussed in reference of the paper by Guo and Rubinstein (Structure of ATP synthase under strain during catalysis. Guo H, Rubinstein JL. Nat Commun. 2022 Apr 25;13(1):2232.). This study studies the yeast ATP synthase with similar goals and methods, but come to some different conclusions. Specifically, they conclude that the central stalk does not distort during ATP hydrolysis and they have a very impressive collection of structures that seem to cover the entire reaction pathway.

We have now included the study by Guo and Rubinstein

“An in-depth study on yeast F₁F₀ ATP synthase⁴⁷, using similar methods to those in this study and which presented the enzyme after incubation with 10 mM ATP, revealed the enzyme in many rotary states. In all structures the rotor was in the same conformation and showed none of the torsional flexing that we observed in the present study, suggesting that the torsional flexing we observe may be limited to the bacterial or *E. coli* enzyme, and may not always be seen with other species.”

REVIEWERS' COMMENTS:

Reviewer #1 (Remarks to the Author):

The criticism of this reviewer has been answered satisfactorily, and I recommend publication of the manuscript.

Reviewer #2 (Remarks to the Author):

The authors have adequately addressed the concerns of the reviewers. This manuscript makes excellent contributions to the understanding of the F1Fo rotary motor mechanism and should be published.

Reviewer #3 (Remarks to the Author):

This is a revised manuscript in which the conclusions have pretty much remained unchanged. The studies investigate the structural changes in the E. coli ATP synthase in the transition from the inhibited state to the active state with the addition of Mg:ATP. A key finding is that the transition to the active state is associated with considerable torsional flexing of the central stalk. An impetus of this study is to understand how the enzyme manages the torsional stress that is a result of the mismatch between the number of protons, 10, that are used for the synthesis of 3 ATPs and the 3 fold symmetry of the enzyme. Both the central and peripheral stalks have been suggested to store torsional energy during the reaction cycle.

The revised manuscript has addressed all of the comments by the 3 reviewers. Overall, this is a very good study and provides new and important findings.